# Orthodontic Compression Enhances Macrophage M2 Polarization via Histone H3 Hyperacetylation

**DOI:** 10.3390/ijms24043117

**Published:** 2023-02-04

**Authors:** Yao Wang, Sabine Groeger, Jiawen Yong, Sabine Ruf

**Affiliations:** 1Department of Orthodontics, Faculty of Medicine, Justus Liebig University Giessen, 35392 Giessen, Germany; 2Department of Periodontology, Faculty of Medicine, Justus Liebig University Giessen, 35392 Giessen, Germany; 3Stomatology Hospital, School of Stomatology, Zhejiang University School of Medicine, Zhejiang Provincial Clinical Research Center for Oral Diseases, Key Laboratory of Oral Biomedical Research of Zhejiang Province, Cancer Center of Zhejiang University, Hangzhou 310003, China

**Keywords:** compressive force, macrophages, polarization, H3 histone, acetylation, adiponectin

## Abstract

Orthodontic tooth movement is a complex periodontal remodeling process triggered by compression that involves sterile inflammation and immune responses. Macrophages are mechanically sensitive immune cells, but their role in orthodontic tooth movement is unclear. Here, we hypothesize that orthodontic force can activate macrophages, and their activation may be associated with orthodontic root resorption. After force-loading and/or adiponectin application, the migration function of macrophages was tested via scratch assay, and *Nos2*, *Il1b*, *Arg1*, *Il10*, *ApoE*, and *Saa3* expression levels were detected using qRT-PCR. Furthermore, H3 histone acetylation was measured using an acetylation detection kit. The specific inhibitor of H3 histone, I-BET762, was deployed to observe its effect on macrophages. In addition, cementoblasts were treated with macrophage-conditioned medium or compression force, and OPG production and cellular migration were measured. We further detected Piezo1 expression in cementoblasts via qRT-PCR and Western-blot, and its effect on the force-induced impairment of cementoblastic functions was also analyzed. Compressive force significantly inhibited macrophage migration. *Nos2* was up-regulated 6 h after force-loading. *Il1b*, *Arg1*, *Il10*, *Saa3*, and *ApoE* increased after 24 h. Meanwhile, higher H3 histone acetylation was detected in the macrophages subjected to compression, and I-BET762 dampened the expression of M2 polarization markers (*Arg1* and *Il10*). Lastly, even though the activated macrophage-conditioned medium showed no effect on cementoblasts, compressive force directly impaired cementoblastic function by enhancing mechanoreceptor Piezo1. Compressive force activates macrophages; specifically, it causes M2 polarization via H3 histone acetylation in the late stage. Compression-induced orthodontic root resorption is macrophage-independent, but it involves the activation of mechanoreceptor Piezo1.

## 1. Introduction

During orthodontic tooth movement (OTM), the compressive force induces fluid strain and hypoxia in the periodontal ligament (PDL), which leads to acute sterile inflammation and subsequent alveolar bone remodeling [1]. Various immune cells, such as macrophages, leukocytes, and monocytes, are recruited to the PDL and participate in the acute inflammation [2]. Generally, it turns into a chronic process within several days [3]. However, the effect of inflammation during OTM may also be adverse when non-regulated or excessive pro-inflammatory cytokine secretion also provokes tissue remolding in the cementum and dentin instead of being limited to bone and periodontal tissue [2,4]. Fortunately, slight root damage is reversible because cementoblasts repair it by producing restorative cementum [5]. However, this repair mechanism is ineffective in 1–5% of patients, which leads to serious external apical root resorption (EARR), a common complication in orthodontic treatment that may lead to tooth hypermobility and, eventually, loss [6]. At present, there are no preventive measures and no therapies for EARR due to its unclear pathogenesis [7]. Recent research has observed a positive correlation between the pro-inflammatory polarization of macrophages and EAAR [8,9]. Further research is necessary to investigate the exact mechanisms.

A great number of reports have indicated biomechanical and biological responses of periodontal ligament cells (PDLCs) and osteocytes during OTM [10,11,12]. However, the direct effects of orthodontic force on macrophages remain unclear, even though their mechanical sensitivity has been confirmed for many years [13,14]. Obviously, investigating the role of macrophages in orthodontic treatment is of high significance for improving clinical strategies. Illustration of the mechanisms in macrophages’ responses to pressure contributes to our understanding of the various pathological changes associated with mechanical stress, such as periodontal destruction resulting from occlusal force [15], lung inflammation caused by excessive mechanical stress [16], and arteriosclerosis related to stagnant blood flow [17].

Macrophages have two different phenotypes in innate immunity: M1 macrophages are mainly involved in pro-inflammatory responses and M2 macrophages are mainly involved in anti-inflammatory responses [18]. The upregulation of macrophage M1 polarization in periodontal tissue was observed in orthodontic force-induced root resorption [8,9]. Thus, it is presumed that compressive force facilitates M1 polarization and indirectly dampens cementum repair. In previous studies, our study group analyzed the influence of adipokines, including leptin and adiponectin, on cementoblasts under compressive force stimulation, since adipose individuals showed worse outcome of orthodontic treatment [19]. In addition, adiponectin was reported to promote M2 polarization [20] and attenuate inflammation [21], and prevents tooth movement. Therefore, we assume it can intervene in the effects of compressive force on macrophages. Bioinformatic analysis was utilized to search for its action target. Furthermore, macrophages can receive mechanical signals through multiple mechanisms [22]. A recent report revealed that H3 histone acetylation is regulated by the rough matrix, which integrates mechanosensation with the inflammatory response in macrophages [23]. Thus, we hypothesized that compression also acts on H3 histone acetylation to induce polarization.

In conclusion, this research investigated mechanisms by which macrophages respond to orthodontic force, and the relevance of this response for orthodontic root resorption.

## 2. Results

### 2.1. Compressive Force Inhibited Macrophages Migration

The macrophages were treated with 1 g/cm^2^ compressive force or adiponectin at different concentrations. After 24 h, the macrophages did not alter morphologically in any of the groups (Figure 1a). The scratch assay results showed that 1 g/cm^2^ compressive force decreased the macrophages’ migration distance (Figure 1b,c), but there was no detectable effect of adiponectin even though it was applied at concentrations up to 10 μg/mL (Figure 1b,d).

### 2.2. Compressive Force Promotes Macrophage Expression of Polarization Markers

After 1 g/cm^2^ compressive force application, inducible nitric oxide synthases (*Nos2*), interleukin 1*β* (*Il1b*), interleukin 10 (*Il10*), and Arginase 1 (*Arg1*) were detected by RT-qPCR. *Nos2*, *Il10*, and *Arg1* gene expressions were all elevated. However, it is noticeable that *Nos2* was the first marker to respond to mechanical compression, showing an increased trend from 6 h (Figure 2a). All the other markers, *Il1b*, *Arg1*, and *Il10*, were enhanced after 24 h (Figure 2b–d). To illuminate the effect of adiponectin on force-activated macrophages, the macrophages were treated with a compressive force and/or 10 μg/mL adiponectin for 24 h before being collected and analyzed. Surprisingly, the results indicated that adiponectin had no effect on any of the markers, whether they were with or without compression (Figure 2e–h).

### 2.3. Compressive Force Promoted Saa3 and ApoE in Macrophages

Setting an adjusted *p*-value of <0.01 and |log2 FC (fold change) | of ≥1 as the cutoff criteria, a total of 28 differentially expressed genes (DEGs) were identified. A total of 21 were upregulated and 7 were downregulated (Table 1 & Figure 3a). To confirm the possible action target of adiponectin, adiponectin and 28 proteins corresponding to DEGs were both mapped to the STRING. According to STRING, eight disconnected genes were filtered out, and the PPI network contained 21 nodes (20 DEGs and adiponectin) and 32 edges (Figure 3b). This shows that serum amyloid A3 (Saa3), apolipoprotein E (ApoE), and interleukin 1*β* (IL-1*β*) connected directly to adiponectin. Then, Saa3 and ApoE expression levels were estimated in the genes to verify the conclusion. The qRT-PCR results revealed that compressive force induced macrophages to up-regulate *Saa3* and *ApoE*, but adiponectin had no effect on them (Figure 2).

### 2.4. H3 Histone Acetylation Mediated M2 Polarization Induced by Compression

Compressive stimulation significantly increased H3 histone acetylation levels (Figure 4a). Furthermore, 500 nM I-BET762 was used to prevent the acetylation reader from recognizing histone.The qRT-PCR results indicated that I-BET762 itself did not affect macrophage cytokine expression. However, it significantly dampened the enhancement of *Arg1* and *Il10* by compression (Figure 4f,g). Meanwhile, it had no effect on the up-regulation of *ApoE*, *Saa3*, *Nos2*, and *Il1b* (Figure 4b–e).

### 2.5. Cementum Repair was Impaired by Compression, but Not by Incubation with Macrophage-Conditioned Medium

To further investigate the regulation of cementum by immune cells, force-activated macrophage medium or 1 g/cm^2^ hydrostatic pressure was used to treat cementoblasts for 24 h. Unexpectedly, although cementoblast migration and osteoprotegerin (OPG) were inhibited directly by compression, neither of them reacted to the macrophage medium (Figure 5a,b). In addition, Piezo1 expression increased in cementoblasts after the application of compression (Figure 5c,d). Thus, a specific inhibitor, GSMTX4, was utilized to incubate cementoblasts simultaneously. Compared to the control group, the independent application of GSMTX4 did not trigger the response of cementoblasts but it significantly blunted migration inhibition and OPG reduction caused by compression (Figure 5e,f).

## 3. Discussion

As mechanosensitive cells, macrophages can receive multiple mechanical signals and transform them into biochemical signals which modulate the microenvironment, inflammatory response, and tissue homeostasis [24]. There is plenty of research that uses matrices with different levels of roughness to illuminate how the mechanical environment regulates macrophages. It was reported that macrophages are altered morphologically on micropatterned extracellular matrix; specifically, they elongate themselves [23]. In this condition, macrophages exhibit lower adhesive capability and dampened pro-inflammatory cytokine expression [25]. This morphological alteration is believed to couple mechanical stimulation and downstream functional output by rearranging the actin cytoskeleton, the Golgi complex, and the cation channel receptor TRPM2 [26]. However, compression simulation inhibited the migration function of macrophages, even though no morphological alteration was observed. This could be due to the compression itself, which, in the case of hydrostatic pressure, is uniformly distributed on the cell membrane. It was demonstrated that macrophages sense hydrostatic pressure through a morphological alteration-independent pathway, which is different from sensing matrix roughness.

Furthermore, *Nos2*, *Il1b*, *Arg1*, and *Il10* were detected by qRT-PCR. Inducible nitric oxide synthase (iNOS) and IL-1*β* are classic markers of LPS-induced M1 polarization. Compressive force promotes M1 polarization to amplify inflammation. This result is consistent with the reports by He et al. [9], Fang et al. [27], and Zhang et al. [8]. Generally, Arg1 hydrolyzes arginine to ornithine and urea, which characterize M2 polarization and antagonize the metabolic pathway in which iNOS is involved [28]. In the same way, IL-10 also suppresses *Nos2* transcription [29]. Surprisingly, compression caused an increase in Arg 1 and IL-10 in our study. According to the updated definition, macrophage polarization is heterogeneous and the two phenotypes are consecutive and interconvertible [30]. Thus, we assume that the synchronous up-regulation of M1 and M2 could be the result of this macrophage heterogeneity in the late stage of the immune response induced by compression. This hypothesis is supported indirectly by single-cell sequencing results [31]. Furthermore, iNOS was the first marker to respond to mechanical stimulation, and did so earlier than Arg1 and IL-10. This indicates that macrophages induce inflammation at the early stage of compression but limit the immune response after 24 h. Identically to biochemical substance-caused macrophage activation, M1 promotes inflammation in the acute stage but M2 mediates inflammatory elimination and tissue repairment in the late stage [32,33]. In any case, compression force can promote macrophage polarization.

Furthermore, it has been frequently reported that adiponectin regulates macrophage polarization by acting on AMPK, NF-kB, and PPAR-γ signaling [20,34,35,36,37,38]. However, according to our results, it has no effect on cellular morphology and migration. More interestingly, polarization markers upregulated by force also do not react to it. Moreover, bioanalysis was deployed to investigate if adiponectin affects mechanically activated macrophages. It indicates that *Saa3* and *ApoE* are upregulated under compression and could be associated with adiponectin. Saa3 is an acute-phase protein [39], and ApoE regulates osteoclast activity, lipid metabolism, and inflammation [40,41], but their roles in OTM still remain unknown. Though further qRT-PCR testing demonstrated that *Saa3* and *ApoE* were enhanced by compression, they both showed no reaction to adiponectin. In any case, our research demonstrated that compression, unlike classic biochemical stimulations (LPS, IFN-γ, IL-4, etc.), activates macrophages via an unknown pathway which is independent of the adiponectin receptor. Additionally, our investigation of the mechanism made a significant impact on current understanding of and intervention in OTM and other mechanically associated pathophysiological process.

The role of epigenetics on mechanical sensation has gradually emerged in recent years. Acetylation and deacetylation are essential chemical modifications of histones, which are associated with gene expression or gene silencing [42]. It has been demonstrated that histone 3 acetylation (H3ac) is associated with IL-10 expression in force-stressed PDL fibroblasts [43]. Our research verified that H3ac is significantly elevated by compression force. Jain et al. reported similar conclusion in 2018 [44]. They found that spatial containment inhibits histone deacetylase 3 (HDAC3) levels, which are negatively associated withhistone acetylation. However, this contradicts Veerasubramanian et al.’s report [23], which showed that H3 hyperacetylation was downsized by using micropatterns as mechanical stimuli. These divergent results could be because different mechanical stimuli were used. Micropatterns creates tension on the underside of the cell that is attached to the well, which is accompanied by morphological alteration of the cell. Compressive force is distributed evenly over the upper cell surface and towards the cell nucleus, which is obviously different from tension created by micropatterns.

Furthermore, the bromodomain and extraterminal domain (BET) protein is a well-known H3ac reader that binds to relaxed regions of the chromatin marked by acetylated histones and act as scaffolds on euchromatin, facilitating binding by transcription elongation factors, transcriptional factors, and other coactivators [45]. IBET762 competed with H3ac to bind to the BET protein and significantly inhibited *Il10* and *Arg1* expression. These results indicate that compression promotes histone H3 hyperacetylation to induce anti-inflammatory polarization (Figure 6). Surprisingly, BET inhibition did not affect M1 polarization in our research, even though Nicodeme et al. [46] and Veerasubramanian et al. [23] both observed a significant reduction in the M1 marker *Nos2* induced by I-BET762. It is noticeable that the stimulation in conjunction with I-BET762 in their study was lipopolysaccharide (LPS), rather than compression. Compression-induced inflammation is obviously different from classical LPS-induced inflammation. As mentioned before, the inflammatory reaction during OTM is regarded as sterile inflammation; nonetheless, various cytokines and immune cells are involved. Regarding the mechanisms, LPS belongs to the pathogen-associated molecular patterns (PAMPs), which activate distinct types of pattern recognition receptors (PRR) that are expressed on the cell surface or intracellularly, activating a signaling cascade and leading to the immune responsive reactions of cells [47,48]. This pathway has been fully described for decades. Mechanical force can activate mechanoreceptors on the cell surface, such as Piezo1 and TRPV4, to regulate downstream ion flux or transcription factors (e.g., Yes-associated protein 1) [49,50,51,52]. Thus, even though histone acetylation is involved in both the mechanical and classical pathways, its roles in the two pathways are likely to be divergent.

Surprisingly, even compression significantly influenced macrophages, but its conditioned medium did not exhibit any effect on cementoblasts as we expected. This contradicts the previous observation that M1 polarization is positively correlated to root resorption in mice [9]. We think there are two potential reasons for this contradiction: 1. though M1 was promoted in the early stage, upregulated M2 in the late stage offsets the effect of M1; 2. there are other mechanically sensitive cells, such as PDLCs, involved in root resorption and their contributions were not addressed in our in vitro research. However, cementoblast functional impairment was confirmed under direct compressive stimulation, which indicates that it is also a mechanically sensitive cell like PDLCs and macrophages. Meanwhile, we detected an increased Piezo1 mRNA level in these cementoblasts subjected to compression. The Piezo1 channel is a mechanically gated ion channel which can be activated by cyclical hydrostatic pressure in murine bone marrow-derived macrophages, followed by Ca^2+^ influx and the production of pro-inflammatory mediators [53]. Thus, we hypothesize that Piezo1 mediates mechanical signal transmission in cementoblasts. The specific inhibitor GSMTX4 was employed to treat cementoblasts, and the effect of force was significantly dampened by it. Thus, compression-induced functional impairment is Piezo1-dependent.

## 4. Materials and Methods

### 4.1. Cell Culture and Reagents

The macrophage cell line RAW264.7 was purchased from the ATCC, and the OCCM-30 cementoblast [54] was kindly provided by Prof. M. Somerman (NIH, NIDCR, Bethesda, MD, USA). Briefly, all cells were maintained in DMEM (41965062, Gibco, New York, NY, USA) containing 10% Fetal Bovine Serum (FBS) (10270-106, Gibco) and 1% penicillin/streptomycin (A3160502, Gibco) and incubated in a humidified atmosphere of 5% CO_2_ at 37 °C. The cells were seeded into 6-well plates (657160, Greiner Bio-One, Kremsmünster, Austria).) at a density of 3 × 10^4^ cells/well until confluence. The cells used were between passages 3 and 7. Mouse adiponectin was purchased from Sino Biological Inc. (Beijing, China) (Cat. No: 50636-M08H) with purity > 95% (determined via SDS-PAGE). Histone acetylation inhibitor I-BET 762 and Piezo 1 inhibitor GSMTX4 were purchased from Sigma-Aldrich (St. Louis, MO, USA) (SML1272, SML-3140).

### 4.2. Compressive Force-Loading

A contactless appliance was developed to load hydrostatic pressure on the cells (Figure 7). The macrophages were cultured under compression (1 g/cm^2^) to conduct further analysis. The morphological structure was observed and recorded using an inverted phase-contrast microscope (Leica Microsystems, Wetzlar, Hessen, Germany). The conditioned medium (CM) from macrophages subjected to 1 g/cm^2^ compressive force was collected and stored at −20 °C.

### 4.3. Scratch Assay

Macrophages and cementoblasts were plated in 6-well plates and cultured. Cells were preincubated for 12 h in starvation medium and wounded via scratching using a 100 μL tip. Through this, a cell-free area was created in the center of the cell layer. Afterwards, all non-adherent cells were washed with 1 × PBS (10010023, Thermo-Fisher, Waltham, MA, USA). Wounded-area images were taken immediately after wounding and 12 h after scratching. The wounded cell layers were photographed at ×10 magnification (Leica Microsystems, Wetzlar, Hessen, Germany) and the wound closure area between the cell layer borders was analyzed and calculated over time using Image J 1.53 software (National Institutes of Health and University of Wisconsin, Bethesda, Maryland, USA).

### 4.4. Identification of DEGs and PPI Analysis

The gene expression profile data of GSE186185 were downloaded from the Gene Expression Omnibus (GEO) of NCBI (http://www.ncbi.nlm.nih.gov/geo/ (accessed on 1 March 2022)) based on the platform of GPL17021 Illumina HiSeq 2500 (*Mus musculus*). This database includes transcriptomes of alveolar macrophage samples after force application in a mouse model of tooth movement and without force application (control). The downloaded profile had been preprocessed, which was carried out with background correction and log2 transformation in R (4.1.0). The DEGs in the case samples were screened and compared with control samples using the Linear Models for Microarray data (limma) package in R. The threshold for the DEGs was set as *p*-value <0.01 and |log2 FC (fold change) | ≥ 1. The STRING database demonstrates known and predicted protein–protein interactions (PPIs) from interaction sources, including text mining, experiments, databases, coexpression, neighborhoods, gene fusion, and co-occurrence. To confirm the possible action target of adiponectin, adiponectin and 28 proteins corresponding to DEGs were both mapped to the STRING; then, we selected the medium confidence (0.4) as the minimum required interaction score, hid the disconnected nodes in the network, and set other parameters as default.

### 4.5. Quantitative Real-Time Reverse Transcriptase–Polymerase Chain Reaction (qRT-PCR)

The total RNA was isolated using the ReliaPrep™ RNA Miniprep System (z6011, Promega, Madison, WI, USA). RNA concentrations were measured at 260 nm using a spectrophotometer (Nanodrop2000, Thermo Scientific, Waltham, MA, USA). cDNA was synthesized from 1.0 μg of total RNA using the Verso cDNA Synthesis Kit (AB1453B, Thermo Fisher, Waltham, MA, USA) and synthesis was performed using a CFX96TM System Cycler (Bio-Rad, Hercules, CA, USA).

The SsoAdvancedTM Universal SYBR@ Green Supermix (1723271, Bio-Rad, Hercules, CA, USA) was used in each reaction setup. The primers employed were mouse inducible nitric oxide synthases (*Nos2)* (QT01547980, Qiagen, Venlo, Limburg, The Netherland), interleukin 1*β* (*Il1b)* (QT01048355, Qiagen), Arginase 1 (*Arg1)* (QT00134288, Qiagen), interleukin 10 (*Il10)* (QT00106169, Qiagen), *Saa3* (QT00249823, Qiagen), and *ApoE* (QT01043889, Qiagen). *β*-Actin (QT00095242, Qiagen) was used as a housekeeping gene. Results were analyzed using Bio-Rad CFX Manager 3.1 software.

### 4.6. Protein Extraction and Western Blot Analysis

RIPA buffer (89901, Thermo Scientific) supplied with 3% protease inhibitor (78442, Thermo Scientific) was used for protein extraction. Protein concentrations were measured using a PierceTM BCA Protein Assay Kit (23225, Thermo Scientific) on a direct reading spectrophotometer (DR/2000, HACH). Further, 20 μg protein samples were separated using 10% SDS-PAGE gel via electrophoresis and transferred to a nitrocellulose membrane (1704271, Bio-Rad). The membranes were blocked with 5% non-fat milk (T145.1, ROTH) in 4 °C for 24 h and incubated with the primary antibodies for Piezo 1 (NBP1-78537, Novus biological, Centennial, CO, USA) and GAPDH (G8795, Sigma, St. Louis, MO, USA) at a concentration of 1:1000. The secondary antibodies employed were: Polyclonal Goat Anti-Rabbit (P0448, Dako, Glostrup Kommune, Denmark) and Polyclonal Goat Anti-Mouse (P0447, Dako, Glostrup Kommune, Denmark) at a concentration of 1:2000. The band signals were detected using a ChemiDoc Imaging System (Bio-Rad) utilizing Amersham ECL Western Blotting Detection Reagents (9838243, GE Healthcare, Chicago, IL, USA).

### 4.7. H3 Histone Global Acetylation Levels

Macrophages were plated in 6-well plates (1.0 × 10^6^ cells/well) and treated with 1 g/cm^2^ as previously described. After 24 h, all adherent cells were harvested. Acetylation level detection was performed using a Histone H3 Total Acetylation Detection Fast Kit (ab115124, Abcam, Cambridge, UK), purchased from Abcam. Samples were prepared, quantified, and assayed according to the manufacturer’s protocol. Finally, the obtained yellow color was read using a microplate reader at 450 nm within 2–15 min.

### 4.8. Statistical Analysis

Statistical analyses were performed using GraphPad Prism 9.0 software (GraphPad software, La Jolla, CA, USA). All values are expressed as means ± standard deviation (SD) and were analyzed using one-way ANOVA or a Student’s t-test for unpaired samples to determine the statistically significant differences between groups. Differences were considered statistically significant at a *p* value of < 0.05. All experiments were repeated successfully at least three times.

## 5. Conclusions

Compressive force impairs macrophage migration and significantly enhances polarization marker expression. Specifically, it promotes M2 polarization via H3 histone hyperacetylation in the late stage of the immune response. In addition, though force-activated macrophages have no effect on cementoblasts, compression directly inhibits cementoblast cementogenesis and migration by affecting Piezo1.

## Figures and Tables

**Figure 1 ijms-24-03117-f001:**
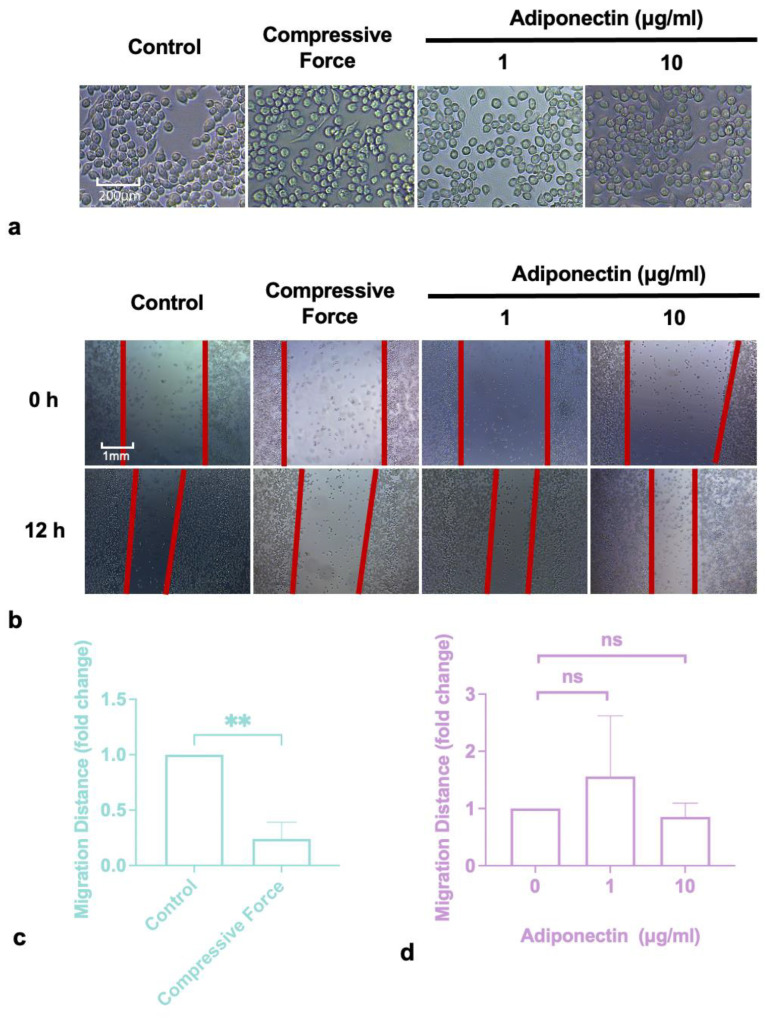
Macrophages were stimulated by the compressive force or adiponectin. (**a**) The effect of compressive force or adiponectin on the morphology of macrophages was negligible. (**b**) The effect of compressive force or adiponectin on the migration of macrophages. (**c**) Compressive force (1 g/cm^2^) significantly inhibited macrophage migration. (**d**) Neither high nor low concentration of adiponectin had an effect on the macrophage migration rate. Values are expressed as means ± SD; ns—not significant; ** *p* < 0.01.

**Figure 2 ijms-24-03117-f002:**
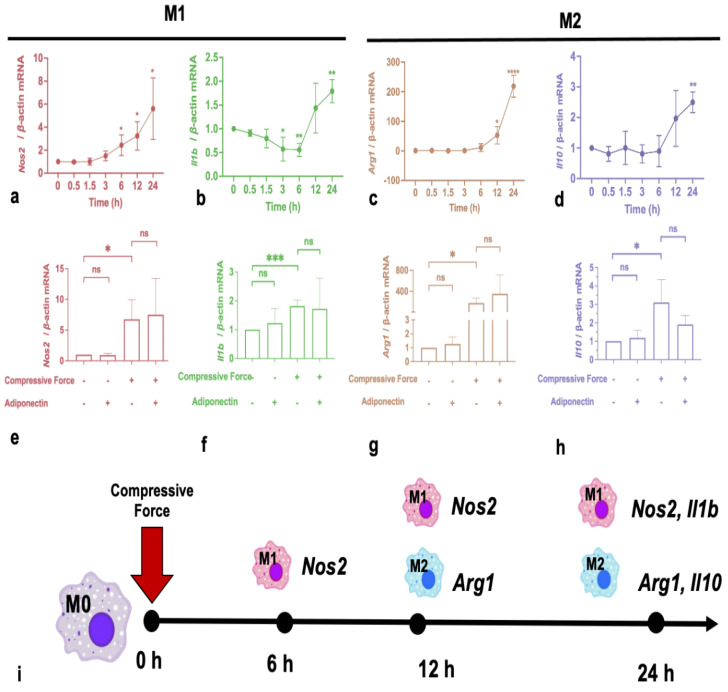
The effects of compressive force on macrophage polarization markers. (**a**–**d**) Compressive force up-regulated *Nos2*, *Il1b*, *Arg1*, and *Il10* levels. (**e**–**h**) Adiponectin did not affect *Nos2*, *Il1b*, *Arg1*, and *Il10* expression. (**i**) *Nos2* increased significantly 6 h after force-loading. *Il1b*, *Arg1*, and *Il10* were elevated after 24 h. Values are expressed as means ± SD; ns—not significant; * *p* < 0.05; ** *p* < 0.01; *** *p* < 0.001; **** *p* < 0.0001.

**Figure 3 ijms-24-03117-f003:**
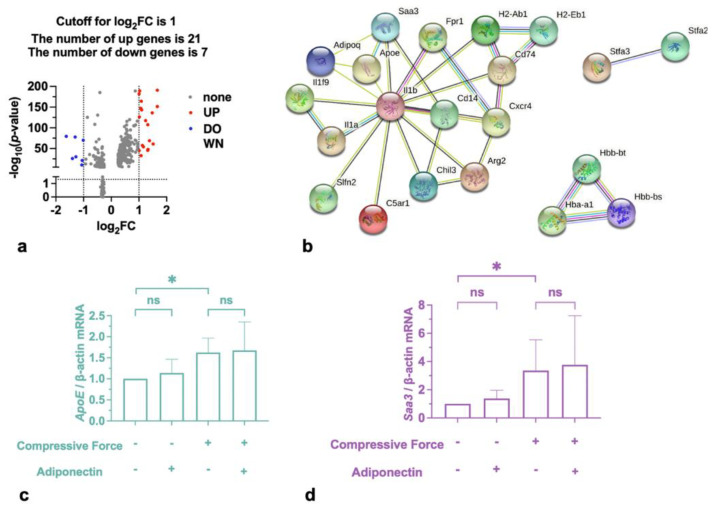
Compression modulates adiponectin-related genes. (**a**) Volcano plot illustrates significant differences in gene expressions pattern between macrophages with and without stimulation by compressive force. (**b**) Based on the STRING online database, top 21 upregulated or downregulated differential genes and adiponectin were filtered into the DEGs PPI network. (**c**,**d**) Increased mRNA expression of *Saa3* and *ApoE* was detected by qRT-PCR. Values are expressed as means ± SD; ns—not significant; * *p* < 0.05.

**Figure 4 ijms-24-03117-f004:**
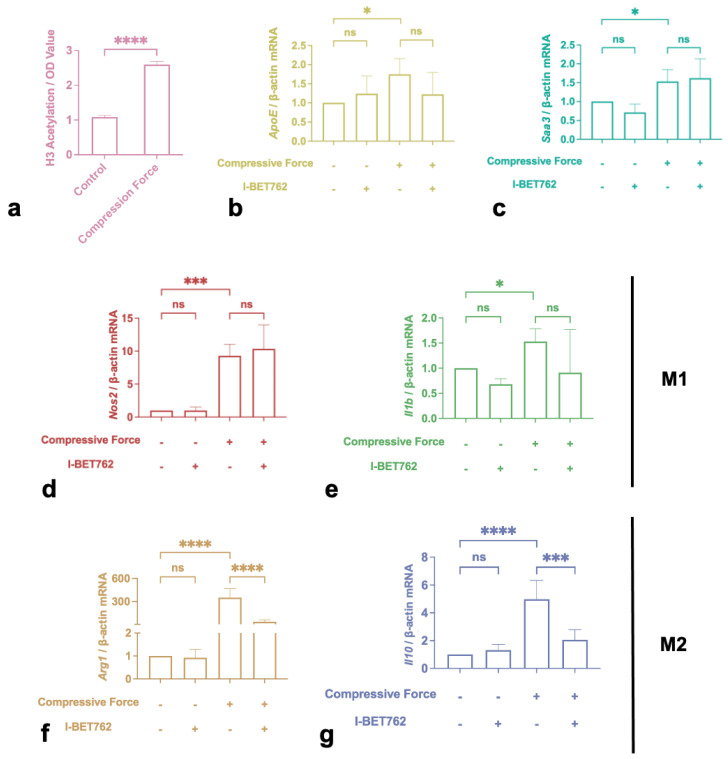
Compressive force induced M2 polarization by promoting H3 histone acetylation. (**a**) H3 histone acetylation level increased after force-loading. (**b**–**e**) *ApoE*, *Saa3*, *Nos2*, and *Il1b* showed no reaction to I-BET762 in the presence or absence of compression. (**f**,**g**) I-BET762 blunted the effects of force on *Arg1* and *Il10* expression. Values are expressed as means ± SD; ns—not significant; * *p* < 0.05; *** *p* < 0.001; **** *p* < 0.0001.

**Figure 5 ijms-24-03117-f005:**
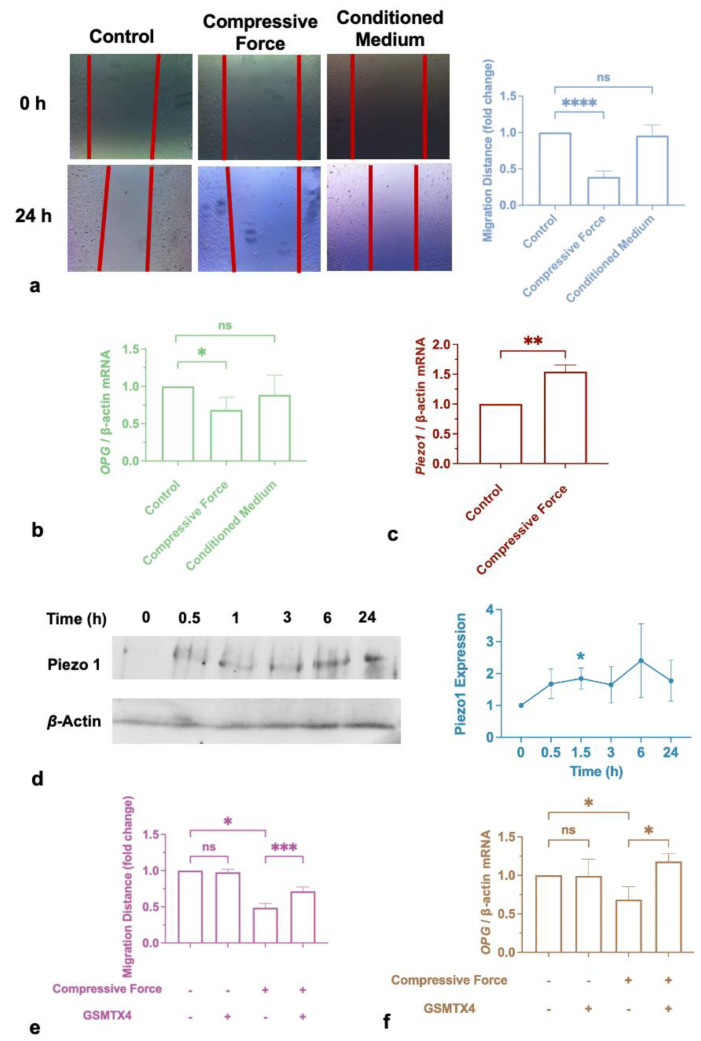
Compression impaired cementoblastic function. (**a**) Migration distance decreased after force-loading in 24 h but did not alter after incubation with conditioned medium. (**b**) qRT-PCR results demonstrate that *OPG* was repressed by compressive force in 24 h. (**c**,**d**) Compressive force triggered cementoblasts to express Piezo1 at the gene and protein levels. (**e**) Piezo1 inhibitor blunted migration impairment induced by compression. (**f**) Piezo1 inhibitor rescued OPG expression in the presence of compression. Values are expressed as sign means ± SD; ns—not significant; * *p* < 0.05, ** *p* < 0.01; *** *p* < 0.001; **** *p* < 0.001.

**Figure 6 ijms-24-03117-f006:**
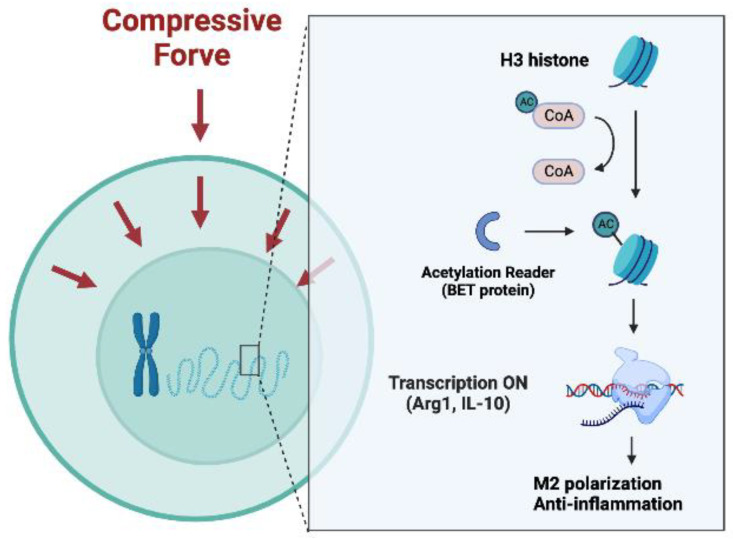
Compressive force enhances H3 histone acetylation to promote M2 polarization.

**Figure 7 ijms-24-03117-f007:**
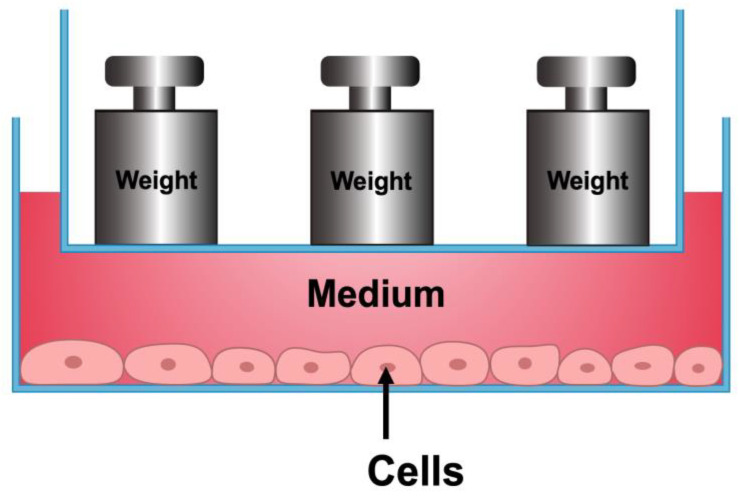
Schematic diagram showing the model used for hydrostatic compressive force. Weight ‘ship’ floats on the medium and compression force is applied to the cells in the form of hydrostatic pressure.

**Table 1 ijms-24-03117-t001:** A total of 28 DEGs were identified in macrophages subjected to force compared to control, including 21 upregulated genes and 7 downregulated genes.

DEGs	Gene Name
Upregulated	*Saa3*, *BC100530*, *Lrg1*, *Chil3*, *Stfa2*, *Il1b*, *Ifitm1*, *Asprv1*, *Fpr1*, *Gm5483*, *Il1a*, *Apoe*, *Il1f9*, *Marcksl1*, *Arg2*, *Stfa2l1*, *C5ar1*, *Cd14*, *Ms4a8a*, *Stfa3*, *Slfn2*
Downregulated	*Cxcr4*, *Cd74*, *H2-Eb1*, *H2-Ab1*, *Hbb-bs*, *Hbb-bt*, *Hba-a1*

## Data Availability

All the data generated in this study appear in the submitted article.

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
