# Peer review of "Orthodontic Compression Enhances Macrophage M2 Polarization via Histone H3 Hyperacetylation"

_ijms, 2023, doi:10.3390/ijms24043117_

Round 1

Reviewer 1 Report

Remarks to the Author:

It is very interesting that compression increases macrophages polarization. However, the results of the article cannot draw relevant conclusions. The following are my comments.

Major points:

1.    Use human macrophage is better. Such THP-1 differentiated macrophage by PMA; PBMC derived macrophage by M-CSF.

2.    mRNA level is too weak to say M1 or M2 phage of macrophage. Detect cell maker like CD80/86, CD163, CD206 by FACS.

3.    Error bar is too high in Fig 5 after I-BET762 treatment. And mRNA changes are small. Dose I-BET762 influence cell viability?

Author Response

JMS-2096753

Orthodontic Compression Enhances Macrophages M2 polarization by Histone H3 Hyperacetylation

Reviewer 1: Remarks to the Author:

It is very interesting that compression increases macrophages polarization. However, the results of the article cannot draw relevant conclusions. The following are my comments.

Major points:

  1. Use human macrophage is better. Such THP-1 differentiated macrophage by PMA; PBMC derived macrophage by M-CSF.

Answer: We show complete new data in this manuscript. The murine model is established in our laboratory from previous work. The usage of a human materials will be subject of future studies.

  1. mRNA level is too weak to say M1 or M2 phage of macrophage. Detect cell maker like CD80/86, CD163, CD206 by FACS.

Answer: Thank you for the suggestion. IL-1β and INOS are both strong markers for M1 polarization as well as Arg and IL-10 are for M2 polarization[1-3] (Bardi et al., 2018; Zhang et al., 2018; Arab pour et al., 2021). The qRT-PCR was also commonly utilized to evaluate polarization state[4, 5] (Chen et al., 2019; Li et al., 2022) and it shows higher sensitivity to polarization of RAW264.7 than proteins assay[2] ( Bardi et al., 2018). Thus, it is approbated to presume the related polarization state from their RNA expression, especially since the expression is distinct and statistically significant.

  1. Error bar is too high in Fig 5 after I-BET762 treatment. And mRNA changes are small. Dose I-BET762 influence cell viability?

Answer: The results were checked for statistical significance. Non-statistically significant results were indicated correctly. The dose of I-BET762 used is 500 nM(line 258), which was referred to the actual s[6, 7].

It is correct. I used 500nM IBET 762 (stated in line 258), this concentration is referred to two literatures. However, neither of them includes cell viability data.

Reviewer 2 Report

This work by Wang et al. explores the possibility of involvement of H3Ac in compressive forces mediated macrophage inflammation relevant to orthodontic tooth movement.

Major comments:

1.    Please include representation images of the scratch assay for Fig 2 and Fig 5, to help the reader visualize the lack of migration in cells experiencing compressive forces.

2.    In the paragraph starting in line 202, describe why the DEG analysis was performed and the PPI network was created. Why was adiponectin treated as an important gene node in the analysis? This needs to be explained better.

3.    The authors do not see reduction in inflammatory genes like Nos2 and Il1b with iBet762 treatment as seen in publications like Nicodeme et al. (PMID: 21068722) and Veerasubramanian et al. (PMID: 34753038). This needs to be explained. Also include an explanation in the discussion as to why compression induced inflammation is different from LPS-induced inflammation.

4.    To implicate H3 acetylation directly to M2 marker activation in compressive force associated macrophage response, chromatin immunoprecipitation (ChIP) assays are need to demonstrate local enhancement of H3Ac at the gene regulatory regions of M2 genes. Similarly, ChIP assays should be used to demonstrate the authors hypothesis that M1 gene activation is not governed by H3 acetylation.

5.    It is of interest to profile the macrophage cytokine secretion induced by compressive forces to see if compressive forces cause them to secrete both M1 and M2 cytokines. This is especially relevant since force-activated macrophage conditioned media did not affect cementoblast migration or Opg expression.

6.    Lines 331-336 need to be rewritten to be more precise with relevant details. Jain et al. (2018) never measured H3Ac levels or M2 markers/cytokines and therefore never directly linked histone acetylation with M2 activation. Similarly, Veerasubramanian et al. (2021) showed that H3 hyperacetylation induced by LPS was downsized by macrophage elongation Macrophage elongation does not cause H3 hypoacetylation in unstimulated cells as suggested by the authors here.

7.    Cementoblasts expressed more Piezo1 mRNA and protein with compression. Does this translate to increased intracellular calcium signals with compressive force application? Additionally, Yoda1 treatment (Piezo1 agonist known to increase calcium signaling) may be explored to see if it causes reduced migration and Opg expression.

Minor comments:

1.    In Fig 1, please also include a photo of the compressive force setup in action to help the readers visualize the experiment better.

2.    In the introduction, include the rationale behind the use of adiponectin (a biochemical cue) in the current work. Why was adiponectin’s effect studied in conjunction with compressive force (a biomechanical cue) is not totally clear.

3.    In Fig 4, the panels c and d have wrong labels for the adiponectin (“- + - +” might have been intended).

4.    When referring to gene expression (mRNA), use standardized italicized notations like Nos2 and Il1b instead of iNos and Il-1β. iNOS and IL-1β indicate the protein, not the gene.

Author Response

JMS-2096753

 Orthodontic Compression Enhances Macrophages M2 polarization by Histone H3 Hyperacetylation

Reviewer 2: Comments and Suggestions for Authors

This work by Wang et al. explores the possibility of involvement of H3Ac in compressive forces mediated macrophage inflammation relevant to orthodontic tooth movement.

Major comments:

  1. Please include representation images of the scratch assay for Fig 2 and Fig 5, to help the reader visualize the lack of migration in cells experiencing compressive forces.
    Answer: The photos were included now.

  1. In the paragraph starting in line 202, describe why the DEG analysis was performed and the PPI network was created. Why was adiponectin treated as an important gene node in the analysis? This needs to be explained better.

Answer: Thanks for this comment. The DEG analysis using STRING database is performed to demonstrates known and predicted protein-protein interactions (PPIs) from interaction sources, including text mining, experiments, databases, coexpression, neighborhood, gene fusion, and co-occurrence. Since adiponectin has been frequently reported to regulate macrophage, we hypothesis it could affect mechanically activated macrophage and searched possible target. Thus, adiponectin was presented as ato observe interactions between them, this was of interest for us, as now described in the introduction and result (line 76-81 &133-134). See minor comments, point 2. This point is mentioned in this part of the manuscript now.

  1. The authors do not see reduction in inflammatory genes like Nos2 and Il1b with iBet762 treatment as seen in publications like Nicodeme et al. (PMID: 21068722) and Veerasubramanian et al. (PMID: 34753038). This needs to be explained. Also include an explanation in the discussion as to why compression induced inflammation is different from LPS-induced inflammation.

Answer: Nicodeme et al.and Veerasubramanian et al used lipopolysaccharide (LPS) to induce Nos2 and Il1b. In our research, these two cytokines were promoted by compressive force. This provides a conclusive explanation for the divergent response to IBET-762. It indicated chemical and mechanical stimulation could exert their effects by totally different signaling pathway.

The compression-induced inflammation is obviously different from classic LPS-induced inflammation. As mentioned before, the inflammatory reaction during OTM is regarded as sterile inflammation and nonetheless various cytokines and immune cells are involved. About the mechanisms, LPS belongs to the pathogen associated molecular patterns (PAMPS) that activate distinct types of pattern recognition receptors (PRR) that are expressed on the cell surface or intracellularly, activating a signaling cascade, leading to immune responsive reactions of cells. This pathway has been fully described for decades. However, mechanisms of mechanosensory are not entirely clear until now, immune cell’s mechanical activation is also an emerging area. Some mechanoreceptors on cell surface have been verified, most of them are mechanical gated ion channels which regulates Ca2+ influx. However, intracellular response to mechanical force is not completely understood. It is what we are going to illuminate in this research.

These aspects have been included in the manuscript now. (Line 416-426)

  1. To implicate H3 acetylation directly to M2 marker activation in compressive force associated macrophage response, chromatin immunoprecipitation (ChIP) assays are need to demonstrate local enhancement of H3Ac at the gene regulatory regions of M2 genes. Similarly, ChIP assays should be used to demonstrate the authors hypothesis that M1 gene activation is not governed by H3 acetylation.

Answer: Chip assays have some disadvantages such as that precipitation can be inefficient, resulting in insufficient protein harvest to test the specificity of the precipitation reaction and that there is a danger that the cross-linking step may fix interactions that are transient and of minor functional significance (Brian Turner, 2001, Cover of Mapping Protein/DNA Interactions by Cross-Linking ; ChIP with Native Chromatin: Advantages and Problems Relative to Methods Using Cross-Linked Material ; Paris: Institut national de la santé et de la recherche médicale ; https://www.ncbi.nlm.nih.gov/books/NBK7099/). In the given aim and experimental setting, we decided that analysis of H3 acetylation together with investigation of specific distinct gene regulations are adequate methods to address the rationale of this study.

Mapping Protein/DNA Interactions by Cross-Linking.

  1. It is of interest to profile the macrophage cytokine secretion induced by compressive forces to see if compressive forces cause them to secrete both M1 and M2 cytokines. This is especially relevant since force-activated macrophage conditioned media did not affect cementoblast migration or Opg

Answer: Thank you for this helpful suggestion. The data presented in this manuscript are completely novel results. Considering that cementoblast doesn’t respond macrophage medium, we are screening more appropriate cell strains as downstream of macrophage at present. After that, analysis of the cytokine pattern will be conducted and addressed in future studies.

  1. Lines 331-336 need to be rewritten to be more precise with relevant details. Jain et al. (2018) never measured H3Ac levels or M2 markers/cytokines and therefore never directly linked histone acetylation with M2 activation. Similarly, Veerasubramanian et al. (2021) showed that H3 hyperacetylation induced by LPS was downsized by macrophage elongation Macrophage elongation does not cause H3 hypoacetylation in unstimulated cells as suggested by the authors here.

Answer: The whole passage was rewritten and the results of the cited studies were described more precisely. We hope it is clearer now, what we intended to state. (Line 398-402)

  1. Cementoblasts expressed more Piezo1 mRNA and protein with compression. Does this translate to increased intracellular calcium signals with compressive force application? Additionally, Yoda1 treatment (Piezo1 agonist known to increase calcium signaling) may be explored to see if it causes reduced migration and Opg

Answer: Thank you for this suggestion. Calcium signaling was not our primary focus. Analysis of calcium signaling, including Yoda1 treatment, will be addressed in future studies.

Minor comments:

  1. In Fig 1, please also include a photo of the compressive force setup in action to help the readers visualize the experiment better.

Answer: All components of model are non-transparent so interior structure is hard to see. Further explanation was included under figure 1 (line 108-110), now diagram is sufficient to understand.

  1. In the introduction, include the rationale behind the use of adiponectin (a biochemical cue) in the current work. Why was adiponectin’s effect studied in conjunction with compressive force (a biomechanical cue) is not totally clear.

Answer: In previous studies we aimed to assess the influence of adipokines, including leptin and adiponectin on cementoblasts under compressive force stimulation, since adipose individuals showed worse outcome of orthodontic treatment. In addition, adiponectin was reported to promote M2 polarization[8], attenuate inflammation[9] and prevents tooth movement. Therefore, we assume it can intervene the effects of compressive force on macrophage and utilize the bioinformatic analysis to search its action target. This is addressed in the introduction of the manuscript now.

  1. In Fig 4, the panels c and d have wrong labels for the adiponectin (“- + - +” might have been intended).
    Answer: The figure was modified as requested.
  2. When referring to gene expression (mRNA), use standardized italicized notations like Nos2 and Il1b instead of iNos and Il-1β. iNOS and IL-1β indicate the protein, not the gene.
    Answer: This was performed as requested.

[1]          Arabpour M, Saghazadeh A, Rezaei N. Anti-inflammatory and M2 macrophage polarization-promoting effect of mesenchymal stem cell-derived exosomes [J]. Int Immunopharmacol, 2021, 97: 107823.

[2]          Bardi G T, Smith M A, Hood J L. Melanoma exosomes promote mixed M1 and M2 macrophage polarization [J]. Cytokine, 2018, 105: 63-72.

[3]          Zhang B C, Li Z, Xu W, et al. Luteolin alleviates NLRP3 inflammasome activation and directs macrophage polarization in lipopolysaccharide-stimulated RAW264.7 cells [J]. Am J Transl Res, 2018, 10(1): 265-273.

[4]          Chen L, Gao B, Zhang Y, et al. PAR2 promotes M1 macrophage polarization and inflammation via FOXO1 pathway [J]. J Cell Biochem, 2019, 120(6): 9799-9809.

[5]          Li N, Chen J, Geng C, et al. Myoglobin promotes macrophage polarization to M1 type and pyroptosis via the RIG-I/Caspase1/GSDMD signaling pathway in CS-AKI [J]. Cell Death Discov, 2022, 8(1): 90.

[6]          Bandukwala H S, Gagnon J, Togher S, et al. Selective inhibition of CD4+ T-cell cytokine production and autoimmunity by BET protein and c-Myc inhibitors [J]. Proc Natl Acad Sci U S A, 2012, 109(36): 14532-7.

[7]          Wang H, Fu H, Zhu R, et al. BRD4 contributes to LPS-induced macrophage senescence and promotes progression of atherosclerosis-associated lipid uptake [J]. Aging (Albany NY), 2020, 12(10): 9240-9259.

[8]          Xu N, Li X, Weng J, et al. Adiponectin Ameliorates GMH-Induced Brain Injury by Regulating Microglia M1/M2 Polarization Via AdipoR1/APPL1/AMPK/PPARγ Signaling Pathway in Neonatal Rats [J]. Front Immunol, 2022, 13: 873382.

[9]          Rizzo M R, Fasano R, Paolisso G. Adiponectin and Cognitive Decline [J]. Int J Mol Sci, 2020, 21(6).

Now it is included. I also modified our explanation in introduction, please check.

In Veerasubramanian’s paper, I-BET762 and Micropattern are two independent stimulations, both them are used to revoke effect of LPS. Thus, the difference between micropattern and compression seems to be another topic though it is also important. I deleted it but kept explanation in manuscript, please check. (Line 402-406)

Round 2

Reviewer 1 Report

This manuscript can be accepted.

Reviewer 2 Report

The authors have tried to address my concerns.

Minor comment: Figure 7 has a typographical error in the word "compressive force".